# Proposals for Standardization of Intraoperative Facial Nerve Monitoring during Parotid Surgery

**DOI:** 10.3390/diagnostics12102387

**Published:** 2022-09-30

**Authors:** Feng-Yu Chiang, Ching-Feng Lien, Chih-Chun Wang, Chien-Chung Wang, Tzer-Zen Hwang, Yu-Chen Shih, Hsin-Yi Tseng, Che-Wei Wu, Yaw-Chang Huang, Tzu-Yen Huang

**Affiliations:** 1Department of Otolaryngology-Head and Neck Surgery, E-Da Hospital, Kaohsiung 824, Taiwan; 2School of Medicine, College of Medicine, I-Shou University, Kaohsiung 824, Taiwan; 3Department of Otolaryngology, E-Da Cancer Hospital, Kaohsiung 824, Taiwan; 4Department of Otorhinolaryngology-Head and Neck Surgery, International Thyroid Surgery Center, Kaohsiung Medical University Hospital, Faculty of Medicine, College of Medicine, Kaohsiung Medical University, Kaohsiung 807, Taiwan

**Keywords:** facial nerve (FN), facial nerve monitoring (FNM), facial expression, parotid surgery, standardization

## Abstract

Intraoperative facial nerve monitoring (FNM) has been widely accepted as an adjunct during parotid surgery to facilitate identification of the facial nerve (FN) main trunk, dissection of FN branches, confirmation of FN function integrity, detection of FN injury and prognostication of facial expression after tumor resection. Although the use of FNM in parotidectomy is increasing, little uniformity exists in its application from the literature. Thus, not only are the results of many studies difficult to compare but the value of FNM technology is also limited. The article reviews the current literature and proposes our standardized FNM procedures during parotid surgery, such as standards in FNM setup, standards in general anesthesia, standards in FNM procedures and application of stimulus currents, interpretation of electrophysiologic signals and prediction of the facial expression outcome and pre-/post-operative assessment of facial expressions. We hope that the FNM standardized procedures will provide greater uniformity, improve the quality of applications and contribute to future research.

## 1. Introduction

Facial dysfunction after surgery for parotid tumors is a common and serious complication. The incidence of facial dysfunction is reported to be 9% to 66% for transient facial weakness and 0% to 9% for permanent facial weakness [1,2,3,4,5,6,7,8,9,10,11,12]. Facial nerve (FN) injury after parotidectomy can lead to facial asymmetry, difficulty chewing, salivation and corneal ulcers, which can seriously affect a patient’s quality of life and may lead to medical litigation. To minimize the risk of FN injury, identifying the FN main trunk and carefully dissecting the FN branches is the standard procedure for parotid surgery.

Currently, intraoperative facial nerve monitoring (FNM) has been popularly accepted as an adjunct to the standard procedure of visual nerve identification during parotidectomy. FNM helps facilitate identification of the FN main trunk and dissection of the FN branches, validation of FN function and prognostication of facial expression after tumor resection [13,14,15,16,17,18,19,20,21,22]. Although the use of FNM is becoming more widespread, little uniformity exists in FNM application in the literature. For example, two-channel or four-channel electromyographic (EMG) recording electrodes may be used, improper use of neuromuscular blocking agents (NMBAs) can occur during general anesthesia and no standards exist for intraoperative FNM procedures and the postoperative application of EMG signal interpretations. Furthermore, no standard facial grading system is available for the preoperative and postoperative assessment of facial expressions. These deficiencies substantially limit the clinical value of FNM technology and hinder the development of innovative research.

In the development of FNM, applying intraoperative neuromonitoring (IONM) to studies on the recurrent laryngeal nerve (RLN) in thyroid surgery is meaningful. Since 2005, we have used neuromonitoring techniques for the RLN and FN as a routine setting in thyroid and parotid surgery. We first published the standardization of IONM of the RLN in thyroid surgery in 2010 [23] and the standardized procedures have been widely followed and included in international standard guidelines for RLN monitoring in thyroid surgery [24,25,26,27]. Inspired by advances in RLN monitoring over the past decade, this article will review the recent literature on FNM and present our (1) standards in FNM setup, (2) standards in general anesthesia, (3) standards in FNM procedures/application of stimulus currents, (4) interpretation of EMG signals/prediction of the facial expression outcome and (5) pre-/post-operative assessment of facial expressions. This article aims to provide greater uniformity through a standardized FNM procedure, promote the quality of application, allow surgeons to elucidate the mechanisms of FN injury, improve their surgical techniques and contribute to future research on FNM techniques.

## 2. Standards in FNM Setup

In the literature, the FNM technique may be used with a two-channel or four-channel EMG signal recording method during parotid surgery. In a two-channel EMG signal recording method, the needle electrodes are placed in the regions of the orbicularis oculi muscle and orbicularis oris muscle. The method of recording the facial muscle activity of two areas cannot reflect the actual function of the four FN branches and correctly evaluate the regional facial muscle function.

The FNM setup consists of three parts: a recording electrode, stimulating probe and monitoring system.

(1) **Recording electrodes**: In our clinical practice, the four-channel EMG signal recording method is routinely used. Four paired needle electrodes (length, 12.0 mm; diameter, 0.4 mm; Medtronic, Jacksonville, FL, USA) were inserted deeply into the muscles over the lower forehead, infraorbital area, superolateral upper lip and inferolateral lower lip on the ipsilateral side to monitor the activity of four regional muscle groups innervated by the temporal, zygomatic, buccal and marginal mandibular branches of the FN (Figure 1A). By comparing the maximal EMG amplitudes detected in each area of the muscle group before and after FN branch dissection, we can understand the activity of each regional facial muscle and the function of each corresponding FN branch after tumor resection.

(2) **Stimulating probe**: A ball-tip (1.0 mm) monopolar probe (Medtronic Xomed, Jacksonville, FL, USA) was used (Figure 1B). The stimulus currents are used from 0.5 mA to 10 mA, depending on the situation and application.

(3) **Monitor system**: The electrode wires, grounding wires and nerve stimulator are connected to the interface box of the monitoring device (Medtronic Xomed, Jacksonville, FL, USA) after placing the recording electrodes and stimulating probe. The duration of stimulation was set at 100 µs, the event threshold was 100 µV, event capture was at the largest signal and the frequency was 4 Hz (Figure 2B,D).

## 3. Standards in General Anesthesia

Intravenous anesthetics, NMBA and inhalational agents are required during general anesthesia. Intravenous anesthetics (propofol or thiamylal) and inhalational agents (isoflurane or desflurane) have little effect on neural monitoring [24,28], but NMBA has a highly sensitive effect on EMG. The use of NMBA is essential during the induction of general anesthesia to achieve safe endotracheal intubation and prevent laryngeal trauma. However, improper use of NMBA may reduce the amplitude of the evoked EMG signal and may also be a potential cause of a false-negative response during FNM. This will interfere with the quantitative analysis of pre-/post-dissection EMG data and reduce the accuracy of postoperative prediction of facial expression. Therefore, discussing neuromuscular blockade issues with anesthesiologists is crucial before starting FNM in parotid surgery.

To reduce the effect of NMBA on EMG data, several options can be selected during the FNM technique in parotid surgery: (1) no use of NMBA [7], (2) short-acting depolarizing NMBA (succinylcholine, 1–2 mg/kg) [29], (3) a small dose of non-depolarizing NMBA (rocuronium, 0.3 mg/kg or mivacurium, 0.2 mg/kg) [30,31] and (4) a standard dose of rocuronium combined with a reversal agent (sugammadex, 1–2 mg/kg) [32]. Many anesthesiologists cannot accept the regimen of not using NMBA during general anesthesia induction because it may increase the risk of difficulty in intubation and laryngeal trauma [33,34,35]. Succinylcholine is a short-acting depolarizing NMBA suitable for FNM but exerts various potential adverse effects, such as hyperkalemia, arrhythmias, vagus neurological arrest and malignant hyperthermia [36]. Therefore, anesthesiologists prefer to use non-depolarizing NMBA during general anesthesia induction. Regarding RLN monitoring, Lu et al. reported that the EMG amplitudes recovered to baseline within 30 min in the succinylcholine and low-dose rocuronium groups. They recommend that low-dose rocuronium (0.3 mg/kg) be used instead of succinylcholine for general anesthesia induction during neural monitoring of the RLN [29].In our clinical study using low-dose rocuronium (0.3 mg/kg) in monitored parotidectomy, the EMG amplitude elicited from the FN trunk was also stable between pre-dissection and post-dissection of the FN [37]. For rapid intubation and better intubation conditions, a higher dose of rocuronium combined with a reversal agent is also feasible and reliable. Lu et al. [32] compared the effect on EMG amplitudes between parotidectomy patients who received two different general anesthesia induction regimens. One group of patients did not receive NMBA and the other group received rocuronium 0.6 mg/kg in combination with sugammadex 2 mg/kg. No significant difference was found between these groups in the EMG amplitudes elicited from the FN trunk before and after dissection of the FN branches.

## 4. Standards in FNM Procedures/Application of Stimulus Currents

Standard FNM procedures and selection of the appropriate level of stimulation current are critical for safe and comprehensive intraoperative monitoring of FN function. In our clinical practice, stimulus currents from 0.5 to 10 mA will be used in different situations and applications.

**Verification (V, 10 mA)**—After skin flap elevation, we will routinely stimulate the mandibular angle area with a 10 mA shunt stimulus current and elicit an EMG signal. The mandible angle area is where the FN mandibular branch runs most superficially. Before localization and identification of the FN trunk, verification of a well-functioning FNM system is necessary. After verification of the functional FNM system, we will dissect the parotid gland from the mastoid bone and cartilage of the external auditory canal.

If an EMG signal cannot be induced during verification, the current return should be checked first. Two situations are possible depending on whether a normal current return is observed. (1) No current return, indicating an abnormal wire connection or equipment malfunction among the stimulating probe, interface box and monitor system. (2) Normal current return, indicating the stimulating probe and monitor system are working and all wires are well connected to the interface box. Next, the correct use of NMBA will be ensured by consulting anesthesiologists. Finally, the recording electrodes on the face will be confirmed for displacement or proper insertion depth.

**Localization (L, 5 mA)** for the FN trunk—The soft tissue and temporo-parotid fascia over the groove are separated layer by layer and the FN trunk is frequently probed with a current of 5 mA during the surgical procedure (Figure 1B).

During the procedure of parotid tumor resection, the first step is to identify the FN trunk at the site of its emergence through the stylomastoid foramen and then carefully dissect the FN branches from the gland. FNM helps facilitate identification of the FN trunk and dissection of FN branches.

**Pre-dissection FN signal (F_1_, 3****–5 mA)**—When the FN trunk is first identified (Figure 2A), we will perform supramaximal stimulation of the trunk at 3–5 mA and the four elicited EMG signals representing the function of temporal, zygomatic, buccal and marginal mandibular FN branches will be displayed on channels 1, 2, 3 and 4 of the monitoring screen (Figure 2B). The four EMG signals are defined as F_1_ signals and are used as basic reference data before the dissection of FN branches.

**Post-dissection FN signal (F_2_, 3–5 mA)**—After dissecting the FN branches and resecting the parotid tumor (Figure 2C), the same stimulus current (3–5 mA) is applied to the FN trunk and the elicited EMG signals are defined as the F_2_ signal (Figure 2D). Next, the EMG amplitudes on each channel of F_1_ and F_2_ (pre-dissection and post-dissection) signals are compared.

When the EMG amplitude on one channel is significantly reduced, a **neural injury point mapping** procedure will be performed with a stimulus current of **1 mA** from the distal to the proximal end of the exposed FN branch. If it **is difficult to differentiate the FN branch** from a fibrous band or a small vessel intraoperatively, a stimulus current of **0.5 mA** will help to differentiate them. The standards in FNM procedures and application of stimulus currents are shown in Figure 3.

## 5. Interpretation of EMG Signals/Prediction of the Facial Expression Outcome

The amplitude of the elicited EMG signal on each channel is the sum of all differences in the potentials of all active motor units of the regional facial muscle group. The EMG amplitude is related to the number of motor units involved in polarization [38,39] and reflects the function of the regional facial muscle group. By comparing the changes in the EMG amplitudes on each channel between F_1_ and F_2_ signals, we can detect the injury to that FN branch and injury severity and predict the outcome of facial expressions in that area. Unchanged or increased post-dissection EMG amplitudes on one channel may indicate normal FN branch function and normal regional facial expression after surgery. Decreased EMG amplitudes may suggest a nerve function deficit or a decreased number of motor units participating in polarization and may lead to facial dysfunction. A significant amplitude decrease suggests that numerous motor units are not involved in polarization and muscle strength may not be sufficient to maintain symmetric facial expressions. However, the correlation between the reduction ratio of the EMG amplitude after FN injury and sufficient muscle strength required for symmetric facial expression needs further study. Mamelle et al. [22] reported that a low post-dissection to pre-dissection ratio of the maximal response amplitude was associated with early postoperative facial dysfunction. Chiang et al. [37] reported that the occurrence of EMG amplitude decreases was common after FN branch dissection, but regional facial weakness did not occur in the patients with a post-dissection amplitude reduction ratio less than 50% in the corresponding FN branch. However, patients with an amplitude reduction ratio greater than 50% had a high incidence (81%) of facial dysfunction.

In our clinical practice, when the post-dissection amplitude decrease ratio is more than 20% on one channel, we will map the injured area of the corresponding FN branch from the distal to proximal end with 1 mA (Figure 3). Thus, we can clarify where and how the nerve is injured. After understanding the mechanisms of nerve injury, we can improve our surgical techniques and predict the possibility of FN function recovery.

## 6. Pre-/Post-Operative Assessment of Facial Expressions

Standardized FNM procedures should begin with a preoperative assessment of facial expressions and end with a postoperative assessment of facial expressions. However, no unified system exists for grading facial function after parotid surgery, making it impossible to compare the results of the study and creating an obstacle for future FNM studies. In the literature [37,40,41,42,43,44,45,46,47,48,49,50,51,52,53], the various FN function grading systems discussed can be divided into global and regional or subjective and objective. The House-Brackmann and Sunnybrook facial grading systems are most commonly used to assess facial function. However, the House-Brackmann grading system measures global facial function, which is most suitable for lesions or injuries to the FN trunk and is not suitable for patients undergoing parotid surgery. The Sunnybrook facial grading system evaluates six facial movements (eyebrows, eyelids, nasal base, upper lip and lower lip). In addition, the system globally assesses resting symmetry, symmetry of voluntary movements and degree of synkinesis movement, which are too complex to perform postoperatively.

Ideally, the facial function assessment should be highly reproducible, fast and easy to perform, have low interobserver variability and consider the severity of facial dysfunction in individual areas. Many facial muscles control various facial expressions. However, wrinkling foreheads, tightly closing eyes, whistling and a wide smile are the main facial movements and are suitable for assessing facial expressions after parotid surgery (Figure 4). We asked patients to perform these four facial movements rapidly and the facial movements were photographed and video-recorded by a smartphone before and after surgery (Appendix A). The dynamic movements of individual muscle groups over four separate facial regions (forehead, peri-orbital area, upper lip and lower lip controlled by the temporal, zygomatic, buccal and marginal mandibular branches of the FN, respectively) are evaluated using our facial grading system.

According to the different grades of facial movement, each of the four facial areas was scored from 3 to 0 points. Normal facial function is defined as the fully symmetrical dynamic movement of the facial area and is assigned a score of 3. Mild facial dysfunction is defined as slightly asymmetrical dynamic movement, but symmetrical expression can be achieved after completion of the movement and is assigned a score of 2. Moderate facial dysfunction is defined as obvious asymmetrical dynamic movement, but it still has little movement and is assigned a score of 1. Severe facial dysfunction is defined as a complete lack of dynamic movement and is assigned a score of 0. For example, a patient with normal facial function in each of the four facial areas would receive a total score of 12 and be registered as T(3)Z(3)B(3)M(3). For clinical cases, please refer to our previously published article [37].

Our proposed facial grading system photographs and video records the four separate static and dynamic facial movements that can be performed quickly and easily before and after surgery, can be scored subjectively and objectively and can be used for long-term follow-up of facial function recovery.

The proposals for standardization of intraoperative FNM in parotid surgery are summarized in Table 1.

## 7. Conclusions

Standardized FNM procedures include pre- and post-operative video recording of facial expressions, grading of facial function by scoring, FNM setup with four-channel recording electrodes, strategies for the optimal use of NMBA under general anesthesia, FNM procedures (V-L-F_1_-F_2_) to obtain reliable and analyzable EMG signals, EMG signal interpretation (F_1_/F_2_ ratio) and neural injury point mapping procedures. We hope that the FNM standardized procedures will provide greater uniformity, improve the quality of applications, help surgeons elucidate the mechanisms of FN injury and improve their surgical techniques and contribute to future studies of FNM technology.

## Figures and Tables

**Figure 1 diagnostics-12-02387-f001:**
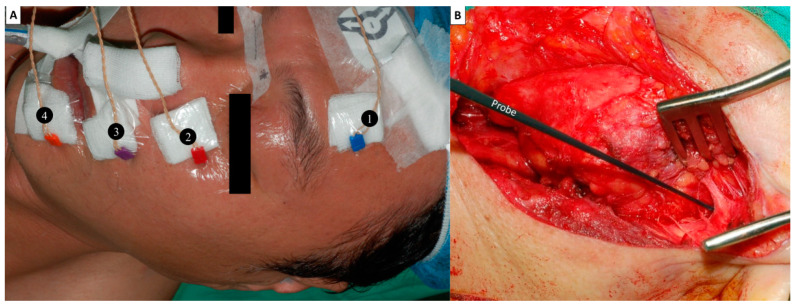
Facial nerve (FN) monitoring setup. (**A**) Recording electrodes: Four paired needle electrodes are inserted deeply into the muscles over the ❶ lower forehead, ❷ infraorbital area, ❸ superolateral upper lip and ❹ inferolateral lower lip on the ipsilateral side to monitor the activity of four regional muscle groups innervated by the temporal, zygomatic, buccal and marginal mandibular branches of the FN. (**B**) A ball-tip (1.0 mm) monopolar stimulating probe was used. The Localization (L, 5 mA) stimulus for the FN trunk was evaluated during parotid gland dissection and facilitated identification of the FN trunk.

**Figure 2 diagnostics-12-02387-f002:**
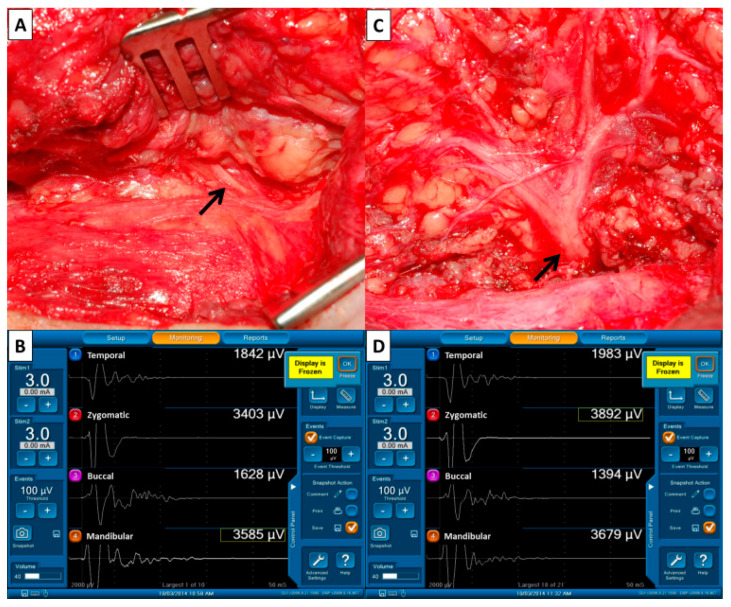
Pre- and post-dissection facial nerve (FN) signals to evaluate FN function. (**A**) The FN trunk (↑) was first identified during parotid gland dissection and we performed supramaximal stimulation of the trunk using 3–5 mA. (**B**) The four elicited EMG signals representing the function of FN branches were displayed on the monitoring screen. The four EMG signals were defined as F_1_ signals and were used as basic reference data before dissection of the FN branches. (**C**) The stimulus current (3–5 mA) was applied to the FN trunk (↑) after dissecting the FN branches and resecting the parotid tumor. (**D**) The four elicited EMG signals were defined as F_2_ signals. The EMG amplitudes on each channel of F_1_ and F_2_ signals were compared.

**Figure 3 diagnostics-12-02387-f003:**
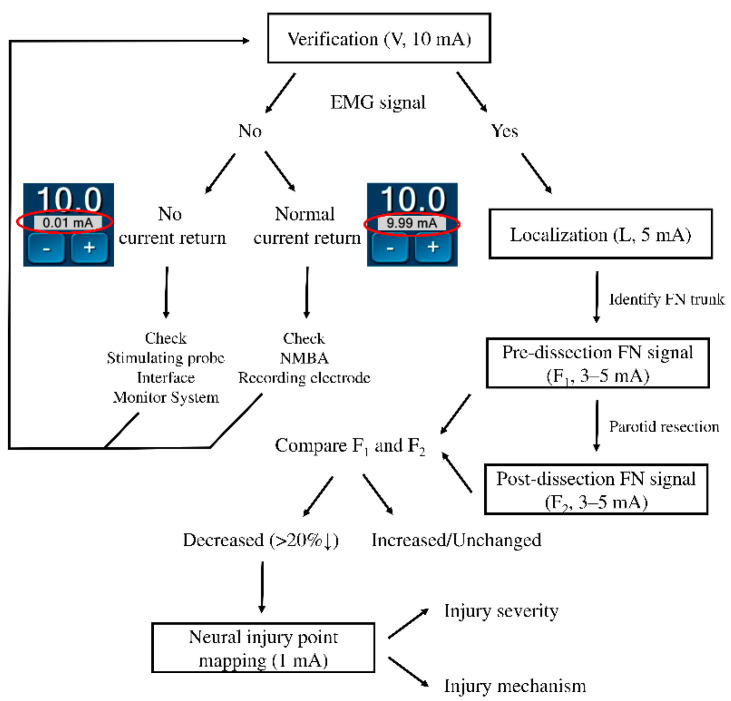
Flowchart of the facial nerve (FN) monitoring procedures and application of stimulus currents. The data in the red circle are the current return during verification with a 10 mA stimulus. EMG = electromyography, NMBA = neuromuscular blocking agents.

**Figure 4 diagnostics-12-02387-f004:**
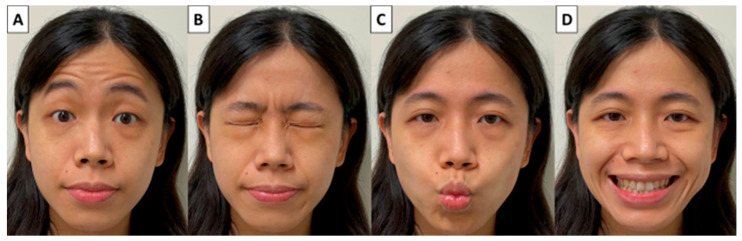
Four main facial movements for assessing facial expressions after parotid surgery. (**A**) Wrinkling foreheads, (**B**) tightly closing eyes, (**C**) whistling and (**D**) a wide smile.

**Table 1 diagnostics-12-02387-t001:** Proposals for standardization of intraoperative FNM in parotid surgery.

Phases	Procedures	Remarks
**Pre-operative**	Pre-operative video recording of facial expression	Four dynamic facial movements: wrinkling foreheads, tightly closing eyes, whistling and a wide smile
**Intra-operative**	FNM setup	Recording electrodes are inserted into the muscles over lower forehead, infraorbital area, superolateral upper lip and inferolateral lower lip
	General anesthesia (NMBA use)	(1) A small dose of non-depolarizing NMBA (rocuronium, 0.3 mg/kg or mivacurium, 0.2 mg/kg), (2) standard dose of rocuronium combined with reversal agent (sugammadex, 1–2 mg/kg)
	FNM procedures (V-L-F_1_-F_2_)	
	V—Verification (10 mA)	Stimulus for mandibular angle area and verification of the functional FNM system before parotid gland dissection
	L—Localization (5 mA)	Stimulus for FN trunk during parotid gland dissection and facilitate the identification of the FN trunk
	F_1_—Pre-dissection FN signal (3–5 mA)	Signals were obtained when the FN trunk is first identified
	F_2_—Post-dissection FN signal (3–5 mA)	Signals were obtained after dissecting the FN branches and resecting the parotid tumor
	Interpretation of EMG signals (F_1_/F_2_ ratio)	Prediction of facial expression outcome
	Unchanged or Increased	Intact FN branch function
	Decreased (>20%↓)	Neural injury point mapping procedure—Mapping the injured area of the corresponding FN branch from the distal to proximal end with 1 mA
**Post-operative**	Post-operative video recording of facial expression	The dynamic movements of individual muscle groups over four separate facial regions are recorded and the total score of the patient’s facial function was 12, registered as T(0–3)Z(0–3)B(0–3)M(0–3)
	0 (Severe facial dysfunction)	A complete lack of dynamic movement
	1 (Moderate facial dysfunction)	Obvious asymmetrical dynamic movement, but movement is observable
	2 (Mild facial dysfunction)	Slightly asymmetrical dynamic movement, but symmetrical expression can be achieved after the movement is completed
	3 (Normal facial function)	Fully symmetrical dynamic movement of the facial area

FN: facial nerve; FNM: facial nerve monitoring; EMG: electromyography; NMBA: neuromuscular blocking agents.

## Data Availability

Not applicable.

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
