# Peer review of "Proposals for Standardization of Intraoperative Facial Nerve Monitoring during Parotid Surgery"

_diagnostics, 2022, doi:10.3390/diagnostics12102387_

Round 1

Reviewer 1 Report

The article discusses the standardization of facial nerve monitoring before, during, and after parotid surgery. They suggest using pre and post-operative assessment of facial movement and facial nerve monitoring during the surgery, with some recommendations for general anesthesia. 

Minor recommendations. 

1.     In Figure 1, panel A, I would recommend increasing the font size for the 1-4 in the figure it is small and difficult to see. 

2.     Figure 3 for the “Check stimulating probe” and Check NMBA” you may want to show an arrow that comes back up to the “Verification” box, indicating after checking those parameters, they should retest the verification. 

3.     Figure 4. Do you have examples of a score of 2, 1, and 0 for the four facial movements? 

Author Response

Dear Reviewer,  

We deeply appreciate your comments.

We have revised our manuscript in-line with the comment made.

The followings are our response:

Response to the Reviewer #1

The article discusses the standardization of facial nerve monitoring before, during, and after parotid surgery. They suggest using pre and post-operative assessment of facial movement and facial nerve monitoring during the surgery, with some recommendations for general anesthesia.

Minor recommendations.

Comment-1:

In Figure 1, panel A, I would recommend increasing the font size for the 1-4 in the figure it is small and difficult to see.

Response:

Thank you for your precious suggestion, the Figure 1 has been modified.

Comment-2:

Figure 3 for the “Check stimulating probe” and Check NMBA” you may want to show an arrow that comes back up to the “Verification” box, indicating after checking those parameters, they should retest the verification.

Response:

Thank you for your precious suggestion, the Figure 3 has been modified.

Comment-3:

Figure 4. Do you have examples of a score of 2, 1, and 0 for the four facial movements?

Response:

Thank you for your comment. In this review article, we avoid the use of identifiable pictures or videos of patients. We add the description “For clinical cases, please refer to our previously published article [37].” in Line 268-269 so that the readers can find a clinical example. We totally agree that more clinical cases should be presented to readers. In fact, we are recruiting patients to establish a reliability and validity study of this scoring system, and expect to provide videos collected from patients with complete informed consent. We believe that the dynamic assessment of facial express will replace the common static assessment methods. Thank you again for mentioning this important issue.

Reviewer 2 Report

The authors provide an overview of the application of neuromonitoring in parotid surgery. The individual steps are described clearly and in detail. From this, it is easy to understand how it works in the clinical setting.

The authors write that they have been using intraoperative neuromonitoring since 2005. Therefore, it would be helpful if their own results were reported: Especially the intraoperative measurements after resection compared with the clinical outcome! Unfortunately, these results are missing.

An other critical point is the use of until to 10 mA Stimulation with the monopoly probe. In our setting, we use only 1mA to maximum 2 mA. Its helpful, to explain the strong use.

Author Response

Author's Response

Dear Reviewer,

We deeply appreciate your comments.

We have revised our manuscript in-line with the comment made.

The followings are our response:

Response to the Reviewer #2

The authors provide an overview of the application of neuromonitoring in parotid surgery. The individual steps are described clearly and in detail. From this, it is easy to understand how it works in the clinical setting.

Comment-1:

The authors write that they have been using intraoperative neuromonitoring since 2005. Therefore, it would be helpful if their own results were reported: Especially the intraoperative measurements after resection compared with the clinical outcome! Unfortunately, these results are missing.

Response:

Thank you for your comment. In this review article, we focus on the description of the procedure steps and try not to describe the case information (unlike an original article with IRB approval). We have a previous published article in Ref. 37. We have the description about the results in Line 219-223, and we add the description “For clinical cases, please refer to our previously published article [37].” in Line 268-269 so that the readers can find a clinical example. Thank you again for mentioning this important issue.

Comment-2:

An other critical point is the use of until to 10 mA Stimulation with the monopoly probe. In our setting, we use only 1mA to maximum 2 mA. Its helpful, to explain the strong use.

Response:

Thank you for this previous comment. In this article, we describe how to appropriately use various electrical stimulation currents in each surgical step. Minimizing the current may be feasible in some cases, but there is also an increased risk of false negatives. Because of the high current safety information had been obtained in previous animal and clinical RLN studies, we believe it is safe to perform the same procedure in parotid surgery. We have also verified the safety and stability of facial nerve stimulation with high current in animal experiments, and we look forward to collecting the data for subsequent publication. Thank you again for mentioning this important issue.